# VideoMB: Steering Representations towards Motion Balanced Caption Generation in Vision-Languange Models

## Abstract

There is a trade-off in Large Vision-Language Models (VLMs) between visual feature compression and spatial fidelity that arises from the tokenization of the input video. We show that this often leads to heavy bias towards appearance, often failing to discern or caption moving objects in a video. To address this limitation, we introduce **VideoMB**, a novel framework that reshapes the model's perceptual priorities toward temporal reasoning. VideoMB incorporates cross-attention layers that establish temporal understanding by modeling information flow between consecutive frames. It employs a fine-tuning paradigm for jointly optimizing caption generation with a global matching objective, constraining the learned visual representations to exhibit maximal similarity with their corresponding counterparts across adjacent frames. This approach is computationally efficient and is seamlessly integrated into existing models. Extensive experiments demonstrate that VideoMB significantly improves motion-based captioning accuracy, particularly for challenging scenarios involving small or low-resolution moving objects, while maintaining competitive performance on appearance-focused tasks.

## 1 Introduction

Recent advances in VLMs showcase remarkable progress, delivering exceptional performance while being resource-efficient (Marafioti et al., 2025; Wang et al., 2024a; Chen et al., 2025) . Yet, these models frequently exhibit lack of motion understanding, tending to describe the sum of the appearance in all frames in lieu of event-level reasoning, due to lack of appropriate inter-frame differencing (Du et al., 2025). Particularly, current architectures typically encode video frames separately along the vision path, followed by compressing visual representations into text-space.

This pipeline provides a desirable trade-off between spatiotemporal resolution and computational efficiency. However, once compressed, temporal understanding must rely on low-resolution representations, which struggle to capture temporal variations accurately. When generating captions for visual inputs that inherently require motion understanding, the outputs are often biased toward appearance, favoring visual elements that are clearly identifiable in single frames. However, as shown in Fig. 1, this strategy is challenged by small of blurry objects, and fails to utilize indicative motion patterns. Our aim is to explore how to balance visual representations such that features of moving cues are prioritized. This way, these cues retain sufficient fidelity even after token-based compression to dominate the model's understanding, aiming to reduce redundant scene details.

Prior work for addressing the challenge of temporal-neglect predominantly rely on visual prompting methodologies and temporal fusion techniques. Du et al. (2025); Hong et al. (2025) show that the lack of motion understanding is not inherently a scalability issue. However, these approaches fall short of offering a generalizable solution for embedding motion priors that consistently constrain models to faithfully represent objects in motion. Moreover, they often equate motion with temporal cues such as event-order or attribute-change (Liu et al., 2024b), and focus specifically on fine-grained human actions. We instead seek a generalizable framework that captures diverse, meaningful object motions across varying dynamics, objects, and scales.

Our hypothesis is that VLMs suffer from limited motion understanding due to misaligned visual representations across temporal sequences. As these models prioritize salient visual features within

individual frames, they fail to maintain consistent spatial positioning across time, resulting in poor comprehension of motion dynamics and movement-related captioning errors. This is in turn, provide appearance-biased representation, which when compressed, lose the ability to discern inter-frame changes (Xu et al., 2024).

Following this hypothesis, the framework presented in this work, **VideoMB**, explicitly integrates motion prior by jointly optimizing caption generation with a global matching objective. A new loss is introduced, which encourages the learned visual representations corresponding to the same semantic information across adjacent frames to be similar. The goal of this objective is to enhance the representation of dynamic elements throughout the video sequence by utilizing the spatiotemporal correspondences.

The primary challenge in establishing temporal correspondences, however, stems from VLMs sampling discrete frames from continuous video sequences. This requires understanding information flow across temporally distant frames, particularly when significant motion, occlusions, or out-of-frame content disrupts visual continuity. As a solution, we leverage the recent state-of-the-art point-tracker CoTracker (Karaev et al., 2024), which robustly tracks points throughout long video sequences, even when the continuous visibility of the point is limited, e.g., by occlusions.

To facilitate the new motion-base objective, VideoMB utilizes cross-attention layers between all pairs of consecutive sampled frames, where queries and keys are derived from the previous frame and values from the next, enhancing features of shared elements to be both more noticeable and temporally aligned. This new attention mechanism is then used for both the representation coherence loss, which requires nearby representations of the same visual element to be similar, and the standard captioning loss to prevent drastic deterioration.

Our approach is computationally efficient, can be easily utilized by existing architectures, and requires no additional training data. In our experiments, we demonstrate that by applying VideoMB on pre-trained VLMs, we gain significant and consistent improvement on motion reasoning evaluation benchmarks. These experiments employ two of the most well-adopted and efficient VLMs, InternVL3-1B (Wang et al., 2024b; Zhu et al., 2025) and smolVLM2 (Zhu et al., 2025). To the best of our knowledge this is the first work to propose a generic low-level approach to enhance VLMs' motion reasoning.

## 2 RELATED WORK

**Video Understanding.** Research on applying VLMs to video understanding has surged in recent years. (Hong et al., 2024; Yao et al., 2024; Xu et al., 2024; Wang et al., 2024b; Marafioti et al., 2025; Wang et al., 2024a; Chen et al., 2025). Standard architectures include a visual encoder that extracts frame-level features, a modality alignment layer that maps them into the language model's embedding space, and an LLM backbone that produces the output. Temporal reasoning therefore depends entirely on how frames are encoded and aligned. Since visual embeddings lack inherent temporal understanding, models often do not appropriately balance positional and spatial cues.

**Motion Understanding in Videos.** Evaluating motion video performance of deep learning models has been a long studied field, with tasks spanning action recognition, action localization, video object detection, and many more (Soomro et al., 2012; Idrees et al., 2017; Vu et al., 2018). Recent work have directly addressed the core challenges related to appearance-biased representations of VLMs, and insufficient datasets focused on movement. Architectural modifications are predominantly limited to general temporal fusion and pooling techniques, including notable papers such as Video-LLaMA (Zhang et al., 2023), which adopted QFormer as a specialized connector, PLLaVA (Xu et al., 2024) using adaptive pooling, and Kangaroo (Liu et al., 2024a) leveraging a unified spatial-temporal patchification. Additional explorations went into visual prompting (Du et al., 2025) and finetuning on synthetic data (Doughty et al., 2024).

Most related to our problem, is the work of Xu et al. (2024), who demonstrates that poor captioning largely stems from bias toward high-norm visual features. To address this, they introduced a temporal pooling strategy aimed at smoothing the feature distribution, thereby dampening the influence of extreme activations. However, this approach is inherently limited: it applies a uniform suppression of strong features rather than adapting to motion dynamics, making it less robust in distinguishing which features should be emphasized or diminished.

In the field of depth from video, Lai & Vedaldi (2025) introduce a component which enhances temporal alignment for image-based models. This component accepts as input both spatial embeddings and point tracks received from off-the-shelf model, and establishes long-term space-time correspondence for high-quality results. This method, unlike ours, requires inference-time reliance to tracking obtained from an external model, and is based on a GNN-like information flow, while our work, which is in a completely different domain, rely on a novel mixed-attention model.

In the domain of video generation, a recent work (Shaulov et al., 2025) enhances motion coherence by reducing patch-wise variance across the temporal dimension during sampling. This inference-time method, which effectively reduces motion artifacts, does not account for object correspondence and does not employ attention manipulation.

**Self-Supervised and Contrastive Learning for Temporal Consistency** A complementary line of research focuses on self-supervised objectives designed to encourage temporal consistency in video representations. Early approaches such as TimeContrast (Dwibedi et al., 2019) exploited cycle-consistency between frames to enforce coherent embeddings across time. Similarly, Speed-Net (Benaim et al., 2020) leveraged the prediction of playback speed as a supervisory signal, implicitly promoting sensitivity to motion dynamics. Building on contrastive frameworks, CVRL (Qian et al., 2021) demonstrated that enforcing temporal alignment via contrastive learning can yield strong video representations without labels. More recently, VTHCL (Wang et al., 2022) introduced temporal hard negative sampling to improve discriminability across motion patterns, while Video-MoCo (Pan et al., 2021) adapted momentum contrastive learning to video sequences, capturing both short and long-term dynamics. These works highlight the effectiveness of self-supervised consistency constraints in mitigating temporal drift and motion neglect. Unlike our method, which directly integrates motion priors into VLM finetuning through cross-attention and auxiliary correspondence losses, self-supervised approaches typically operate during pretraining, with weaker alignment to downstream captioning tasks.

## 3 VIDEOMB

VideoMB consists of two complementary contributions (see Fig 2): (i) The integration of cross-attention on consecutive frame embeddings; (ii) Dual-objective loss to align patch embedding features along the temporal dimension, based on their mutual semantic information. During finetuning, we freeze the model's original weights, solely finetuning the cross-attention layers.

### 3.1 FEATURE MATCHING

Along the vision encoder path, we incorporate feature-matching cross-attention layers that extract distinctive features for aligning adjacent frames. Let $F \in \mathbb{R}^{f \times hw \times C}$ denote the input feature map, where $f$ is the number of frames, $h$ and $w$ are the spatial dimensions, and $C$ is the feature dimension. We denote by $F_t \in \mathbb{R}^{hw \times C}$ a slice of $F$ along the temporal dimension.

To incorporate spatial awareness into our frame embeddings, we add a fixed 2D sine and cosine positional encoding $P \in \mathbb{R}^{hw \times C}$ to each frame embedding, following (Carion et al., 2020). This encoding provides spatial context by encoding the original spatial locations from which features were extracted, enabling the model to maintain spatial correspondence across temporal sequences.

For each pair of adjacent (consecutive sampled) frames $F_t$ and $F_{t+1}$, we perform cross-attention where the query $Q$ is derived from the current frame $F_t + P$, while the key $K$ and value $V$ are computed from the subsequent frame $F_{t+1} + P$:

$$Q_t = (F_t + P)W_Q \tag{1}$$
$$K_{t+1} = (F_{t+1} + P)W_K \tag{2}$$
$$V_{t+1} = (F_{t+1} + P)W_V \tag{3}$$

where $W_Q$, $W_K$, and $W_V$ are learned linear projection matrices. The cross-attention mechanism then computes:

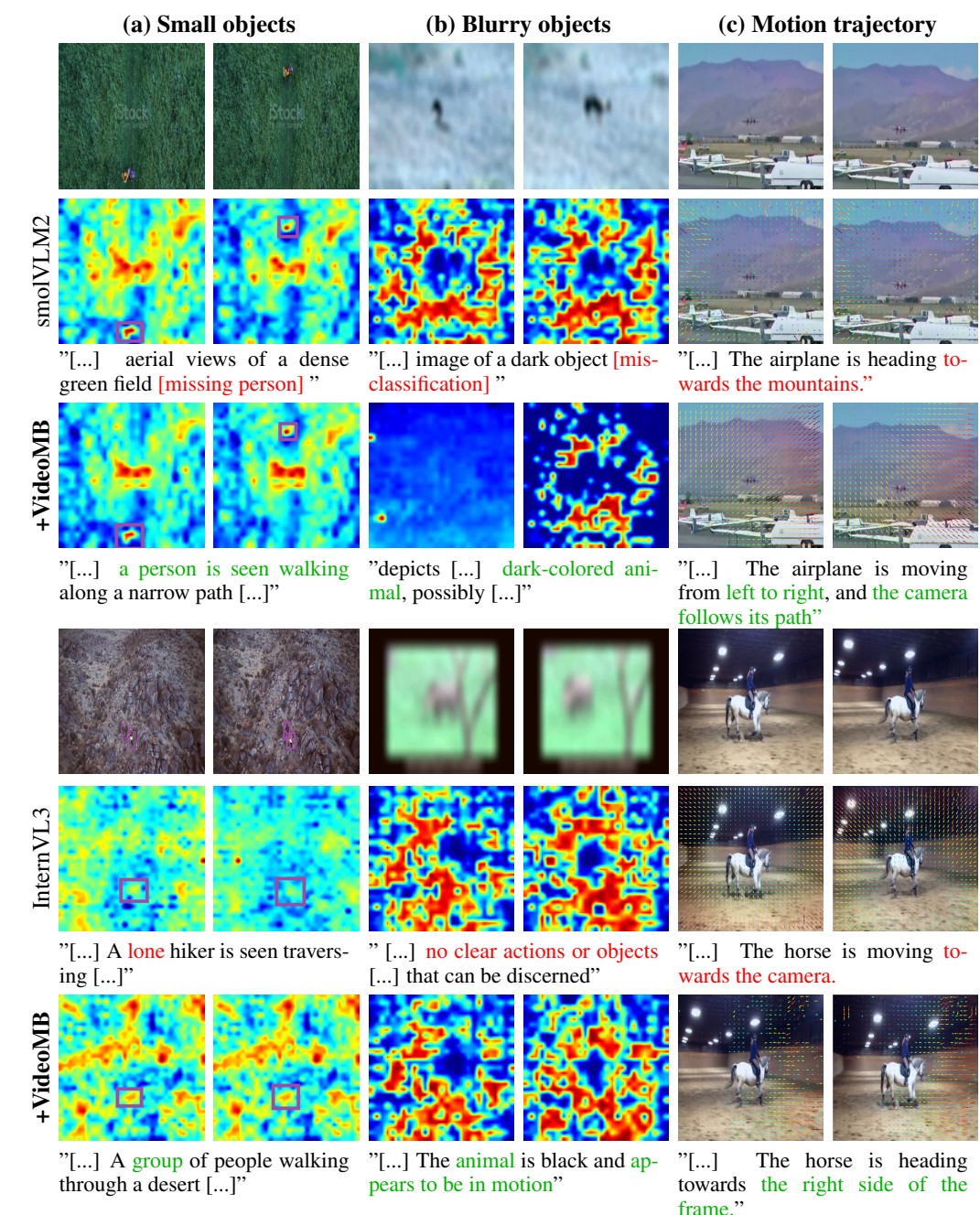

Figure 1: **Qualitative results.** Caption generation results before and after applying VideoMB on (a) small object (b) blurry objects and (c) motion trajectory understanding. For (a) and (b) we visualize the embeddings' norm heatmap, understanding which features recieve the most attention. For (c), we visualize cross-corresponding features, allowing us to understand the inner reasoning behind the model's predictions. (Marafioti et al., 2025).

$$F'_t = \text{Attention}(Q_t, K_{t+1}, V_{t+1}) = \text{softmax}\left(\frac{Q_t K_{t+1}^\top}{\sqrt{C}}\right) V_{t+1} \qquad (4)$$

This cross-attention operation produces enhanced frame embeddings $F'_t \in \mathbb{R}^{hw \times C}$ for all frames except the last one ($t = 1, 2, \ldots, f - 1$). The enhanced embeddings capture motion-relevant fea-

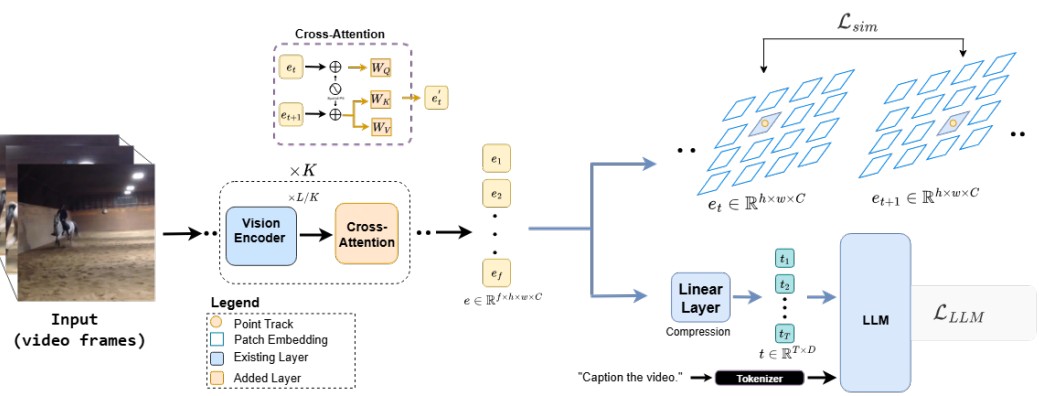

Figure 2: **Overview of VideoMB framework**. We uniformly place $K$ Cross-Attention Layers along the vision encoder path, resulting in feature enhancement of moving objects. Next, we jointly optimize both caption generation using the baseline's initial predictions, and a global matching objective where we compare feature similarities by correlating all pair-wise features of adjacent frames.

tures by attending to spatially corresponding regions in the subsequent frame, effectively learning to identify and match visual patterns that persist or transform across adjacent time steps.

To preserve the original frame information while incorporating the motion-aware enhancements, we employ a residual connection:

$$F_t'' = F_t + F_t' \tag{5}$$

where $F_t''$ represents the final motion-balanced frame embedding. This additive combination ensures that the enhanced representations $F_t''$ retain the essential visual content from the original frame $F_t$ while being augmented with motion-aware features from $F_t'$.

### 3.2 MOTION-BALANCED REPRESENTATIONS

Once passed through the last transformer block, we enforce an auxiliary loss constraint such that corresponding latents in consecutive frame share high similarity. We compare the feature similarity for each embedding in $F_t''$ with respect to all patches in $F_{t+1}''$ by computing their correlations.

This is accomplished by computing a normalized correlation matrix:

$$M = \text{softmax}\left(\frac{F_t'' F_{t+1}''^\top}{\sqrt{C}}\right) \in \mathbb{R}^{h \times w \times h \times w}, \tag{6}$$

which provides a pseudo-probability distribution over patch correspondences between consecutive frames.

Let $G \in \mathbb{R}^{hw \times 2}$ be the matrix that holds in each column $i = 1, \ldots, hw$ the image coordinates of patch $i$. These coordinates are mixed by the attention matrix $M$ to provide displacement $V$, which is computed as the difference between the patch coordinates before and after mixing:

$$V = MG - G \in \mathbb{R}^{hw \times 2}. \tag{7}$$

A supervision signal for $V$ is obtained from the CoTracker network (Karaev et al., 2024), which provides $T = hw$ tracks per frame in the form of the tensor $U \in \mathbb{R}^{f \times T \times 2}$. (each track consists of a sequence of 2D point displacement across $f$ frames). We run CoTracker such that the tracks start at the patch center in the first frame. This tracking model also provides a binary visibility mask $\mathcal{B} \in \{0, 1\}^{f \times T}$, which indicates whether each tracked point is visible (not occluded or out-of-bounds) in each frame.

The auxiliary loss $\mathcal{L}_{sim}$ is computed as the $L_1$ distance between the predicted movement vectors $V$ and the ground truth movement vectors derived from the tracks $\tau$:

$$\mathcal{L}_{sim} = \frac{1}{\sum_{i=1}^{T} \mathcal{B}_i} \sum_{i=1}^{T} \mathcal{B}_i \cdot ||V_i - U_i||_1 \,, \tag{8}$$

where $U_i$, $V_i$, and $B_i$ are slices of the corresponding tensors along the patch index dimension. This loss encourages the learned correspondences to align with the motion patterns observed in the ground truth tracks, while only enforcing constraints on visible pixels that are not occluded or out-of-bounds in the subsequent frame.

Finally, the training objective is computed by the following weighted loss:

$$\mathcal{L} = \alpha \mathcal{L}_{sim} + \beta \mathcal{L}_{LLM} \tag{9}$$

where $\mathcal{L}_{LLM}$ is the typical Cross-entropy loss for language models on the captioning task, applied to the cross-frame-attention features $F''$, and $\alpha$ and $\beta$ are hyper-parameters.

## 4 EXPERIMENTS

We conduct both qualitative and quantitative experiments to demonstrate VideoMB's effectiveness in enhancing temporal reasoning. Our results show improved motion understanding, even under challenging conditions such as blurry videos or small objects, while also enabling visualization of norm values that distinguish success from failure.

**Implementation details**. We employ two of the most well-adopted and efficient VLMs, InternVL3-1B (Zhu et al., 2025) and smolVLM2 (Marafioti et al., 2025). In all of our experiments, we use a learning rate $\eta$=1e-4, leveraging the Adam optimizer, on a single NVIDIA H100 GPU with 70GB memory, for 2700 steps and batch size of 8. The InternVL3-1B and smolVLM2 are evaluated on images at a resolution of 448x448 and 384x384, respectively. For the finetuning we leverage the GOT-10k (Huang et al., 2021) video dataset, which consists of a wide range of moving dynamics, object sizes, and environments. We set $\alpha = 0.1$ and $\beta = 1.0$ for both models.

For quantitative evaluation, we employ two prominent benchmarks for motion reasoning in VLMs: TempCompass (Liu et al., 2024b) and the "sports" subset of MotionBench (Hong et al., 2025). While these represent the current state-of-the-art in motion evaluation datasets, they primarily focus on fine-grained human movements in short video clips with limited motion magnitude. We specifically utilize the sports category of MotionBench as it contains the most substantial object displacement among available categories, providing sufficient motion range to meaningfully test our approach.

### 4.1 QUALITATIVE RESULTS

We evaluate our method's ability to attend to moving elements through three challenging video scenarios: (i) small objects, (ii) blurry objects, and (iii) complex motion trajectories (see Fig. 1). Each case includes visualization to interpret the model's captioning decisions.

**Small objects.** While the baseline model neglects to mention a person moving in the scene, our method successfully does. The patch embeddings heatmap reveals that although both models detect the person's presence (shown by high activation values at the person's location), the baseline's attention is scattered more across the background scenery, preventing proper recognition.

**Blurry Objects.** Our model correctly identifies a blurry object as an animal, whereas the baseline fails to make this determination. Visualization of mid-video and final frame shows distinct attention patterns: the baseline constantly exhibits scattered high-norm values across the background in the final frame, reflecting appearance bias. In contrast, our method concentrates attention more precisely for the last frame, and in mid-video frames, where only their frame embeddings are amplified through cross-attention, high-norm values are highly localized. This demonstrates that our approach consolidates global motion information into concentrated representations.

**Motion trajectory.** The baseline incorrectly predicts an airplane moving toward mountains, while our method accurately captures the true trajectory. The 2D displacement vector visualization perfectly aligns with the model's reasoning, with all vectors pointing to the region of the airplane and

Table 1: **Tempcomp evaluation results.** A comparison of overall temporal understanding, before and after applying VideoMB on smolVLM2 (Marafioti et al., 2025) and InternVL3-1B (Zhu et al., 2025), using the Tempcomp benchmark

| | Tempcomp | | | | |
|---|---|---|---|---|---|
| | Localization | | Event-level Reasoning | | |
| | Direction ↑ | Speed ↑ | Attribute Change ↑ | Action ↑ | Order ↑ |
| SmolVLM2 | 0.47 | 0.50 | 0.57 | 0.85 | **0.49** |
| SmolVLM2+*VideoMB* | **0.51** | **0.52** | **0.61** | **0.86** | 0.42 |
| InternVL3-1B | 0.48 | 0.52 | 0.58 | **0.86** | **0.57** |
| InternVL3-1B +*VideoMB* | **0.50** | **0.54** | **0.60** | 0.79 | 0.47 |

mountains. In contrast, for VideoMB, these vectors can be divided into two directions: the direction the airplane is heading (right) and the rest of the points pointing left (background), as the camera is panning right. This correlates with the model's prediction on which direction the airplane is heading.

Visualizations of feature displacements, which are primary focus of this work, are reported in Fig. 3. Our results demonstrate VideoMB's effectiveness in fundamentally restructuring feature similarities within the latent space, enabling the model to capture motion dynamics for both objects and camera movement. The displacement vectors in VideoMB precisely align with the underlying motion patterns in the video, whereas the baseline exhibits persistent self-similarity artifacts that disrupt temporal coherence at the feature level, as evidenced in Figure 1. This transformation of low-level feature representations directly translates to improved motion understanding capabilities.

### 4.2 QUANTITATIVE RESULTS

The results, presented in Tab. 1,2 demonstrate significant improvements for motion-related benchmarks, all while not finetuning our model with additional data, inherently achieving improvement via a low-level approach.

**Tempcomp.** We evaluated our method on the TempComp benchmark by categorizing questions into two groups: (i) Spatial-Temporal Localization (speed and direction queries) and (ii) Event-Level Reasoning. For smolVLM2, our approach achieved substantial improvements in Spatial-Temporal Localization, with gains of 4%+ for **direction** and 2%+ for **speed** recognition. We report more modest improvements of 2% for the two categories. For Event-Level Reasoning, we observed a notable 4%+ improvement in **attribute change** detection for smolVLM2, and 2% for InternVL3, as well as a small gain of (1%) in **action** recognition tasks for smolVLM2. For order-related questions involving fine-grained movements, performance was limited for both models, due to the prevalence of small-range motions in the dataset. The fact that we are able to achieve significant improvement in this metric for Motion Bench, supports this claim.

**Motion Bench.** we recorded consistent improvements for both models with the exception of Motion-related objects detection. Most noticeably, we achieve a significant 12% boost for smolVLM2 and 16% for InternVL in **repetition count**, 6% for **action order** and 3% gain for **motion recognition** for smolVLM2. For motion-related objects, our method showed showed constrained improvements, as many queries focused on basic object classification (e.g., distinguishing volleyball from basketball).

The improvements observed for smolVLM2 were overall substantially more significant than those for internVL3, which can be attributed to smolVLM2's higher compression ratio (4:1 vs 2:1) resulting in greater spatial information loss that our approach specifically addresses. The striking parallel trends across both models, such as significant improvements in repetition count on Motion Bench yet poor performance on the Order metric in Tempcomp, suggest that dataset characteristics also influence model performance.

### 4.3 ABLATION STUDY

We ablate the primary design choices to disentangle their individual contributions. (i) First, we explore how fine-tuning the model only with the captioning loss, without the feature-alignment supervision, affects performance. (ii) Second, we test whether temporal correspondence is essential

Table 2: **Motion Bench (sports subset).** A comparison of overall temporal understanding before and after applying VideoMB onon smolVLM2 (Marafioti et al., 2025) and InternVL3-1B (Zhu et al., 2025), using the "sports" subset of MotionBench benchmark.

| | Action Order ↑ | Motion Recognition ↑ | Motion-related Objects ↑ | Repetition Count ↑ |
|---|---|---|---|---|
| **Motion Bench (sports subset)** | | | | |
| SmolVLM2 | 0.28 | 0.47 | **0.86** | 0.12 |
| SmolVLM2+*VideoMB* | **0.34** | **0.50** | 0.80 | **0.24** |
| InternVL3 | 0.33 | 0.50 | 0.82 | 0.36 |
| InternVL3+*VideoMB* | **0.34** | **0.52** | **0.84** | **0.52** |

Figure 3: **Cross-Correspondence Qualitative Results.** Visualization of cross-similarity between adjacent frames, before and after applying VideoMB on SmolVLM2.

by replacing our cross-frame similarity loss, with an alternative loss which enforces similarity between per-frame patches and the moving object. This comparison isolates whether the gain comes from capturing inter-frame consistency or object-centric emphasis. (iii) Third, we study the effect of masking in the auxiliary loss, clarifying the importance of selectively applying temporal constraints versus enforcing them uniformly.

Table 3: **Ablation Study**. Ablations of VideoMB include: (i) removing the cross-frame alignment loss (w/o alignment), (ii) replacing it with a per-frame maximal-similarity objective (w/o cross-alignment), and (iii) removing the binary mask signaling visible displacements (w/o masking). We further evaluate the effect of the cross-alignment hyperparameter weight loss $\alpha$.

| | Tempcomp | | | | |
|---|---|---|---|---|---|
| | Localization | | Event-level Reasoning | | |
| | Direction ↑ | Speed ↑ | Attribute Change ↑ | Action ↑ | Order ↑ |
| w/o alignment | 0.48 | 0.51 | 0.58 | 0.81 | **0.48** |
| w/o cross-alignment | 0.50 | 0.51 | 0.57 | 0.78 | 0.41 |
| w/o masking | 0.46 | 0.47 | 0.56 | 0.85 | 0.41 |
| VideoMB ($\alpha = 0.05$) | 0.49 | **0.53** | 0.61 | 0.85 | 0.47 |
| VideoMB ($\alpha = 0.5$) | 0.50 | 0.52 | 0.59 | 0.81 | 0.44 |
| VideoMB ($\alpha = 1.0$) | 0.48 | 0.52 | 0.57 | 0.80 | 0.43 |
| VideoMB ($\alpha = 5.0$) | 0.48 | 0.51 | 0.57 | 0.75 | 0.43 |
| VideoMB (ours, $\alpha = 0.1$) | **0.51** | 0.52 | **0.61** | **0.86** | 0.42 |
| | Motion Bench (sports subset) | | | |
| | Action Order ↑ | Motion Recognition ↑ | Motion-related Objects ↑ | Repetition Count ↑ |
| w/o alignment | 0.30 | 0.45 | 0.80 | 0.18 |
| w/o cross-align | 0.27 | 0.44 | 0.65 | 0.12 |
| w/o masking | 0.29 | 0.48 | 0.69 | 0.14 |
| VideoMB ($\alpha = 0.05$) | 0.32 | 0.48 | 0.80 | 0.21 |
| VideoMB ($\alpha = 0.5$) | 0.32 | 0.45 | 0.80 | 0.12 |
| VideoMB ($\alpha = 1.0$) | 0.32 | 0.47 | **0.81** | **0.30** |
| VideoMB ($\alpha = 5.0$) | 0.30 | 0.44 | 0.80 | 0.30 |
| VideoMB (ours, $\alpha = 0.1$) | **0.34** | **0.50** | 0.80 | 0.24 |

Moreover. we examine how varying the weighting hyperparameters of the auxiliary loss influences the outcome.

Quantitative results of the ablation study on SmolVLM2 are reported in Tab. 3. Evidently, ablating the core components of our method leads to a substantial overall decrease in performance. First, removing the cross-alignment loss leads to worse performance than our full method. These results demonstrate that the cross-attention mechanism alone does not inherently improve temporal reasoning. This outcome aligns with our expectations: training solely on the captioning objective drives the model toward a limited learning space, potentially degrading its capabilities and failing to provide any explicit inductive bias for motion understanding. We further report that removing masking drops the accuracy of the localization-related reasoning for Tempcomp, as well as an overall decrease on Motion Bench. This is reasonable, since training forces the model to predict semantic correspondences that do not exist, making the entire training process less stable. A more modest decline in localization reasoning for Tempcomp is observed when ablating alignment. It stands to reason that constraining the model to pay more attention to objects facilitates these tasks; however, doing so introduces a clear trade-off with temporal understanding, as per-frame localization becomes too dominant at the expense of a robust understanding of motion dynamics. Finally, varying the cross-alignment hyperparameter weight loss yields similar results overall, with observable gains in some settings, but our chosen value remains the best.

## 5 LIMITATIONS

Our method improves motion coherence and temporal understanding but has several limitations. First, there is misalignment between point track's spatial resolution and the patch embeddings' resolution (downsampled) which may result in incomplete feature correspondence supervision. While extensive training data should promote general understanding, extremely small objects may still pose tracking challenges. Moreover, VideoMB inherits the constraints of its pre-trained foundation models. The enhancement layers can only amplify features that the base model already attends to, limiting improvements for small or blurry elements despite fine-tuning on large datasets. Second, current benchmarks inadequately evaluate our model's capabilities, as most focus on subtle motion

cues and human actions rather than diverse visual scenarios. We hope this work inspires development of VLMs for broader visual understanding tasks, e.g. aerial scene interpretation.

## 6 CONCLUSIONS

We introduced VideoMB, a novel framework that addresses the fundamental trade-off between visual compression and spatial fidelity in VLMs, demonstrating a generic approach to reshape perceptual priorities toward temporal reasoning. Through our cross-attention mechanisms and joint optimization of caption generation with global matching objectives, our approach enables models achieve consistent improvements on motion-related tasks, especially for challenging scenarios involving small or low-resolution moving objects. VideoMB's computational efficiency and seamless integration into existing architectures make it a practical solution for enhancing temporal understanding in video analysis. Future work will explore extending these temporal reasoning capabilities to longer video sequences and investigating applications in other video understanding tasks beyond captioning, potentially opening new avenues for more comprehensive multimodal video analysis systems.

LARGE LANGUAGE MODEL USAGE

We used large language models (ChatGPT/GPT-4, Claude) to assist with writing, editing, and proof-reading portions of this manuscript. All technical content, experimental design, implementation, and scientific contributions remain entirely our own work. The models were not used for data analysis, result interpretation, or generating scientific claims.

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

## A FAILURE CASES - DEPENDENCY ON EXTERNAL TRACKER

Our method relies on pseudo-labels generated by the external point tracker CoTracker3 (Karaev et al., 2024). Below we present qualitative examples of failure cases originating from its limitations. As CoTracker3 often struggles to track points on featureless surfaces, it produces noisy displacement vectors, leading our model to exhibit flawed motion pattern understanding in such scenarios.

**CoTracker**  **smolVLM2**

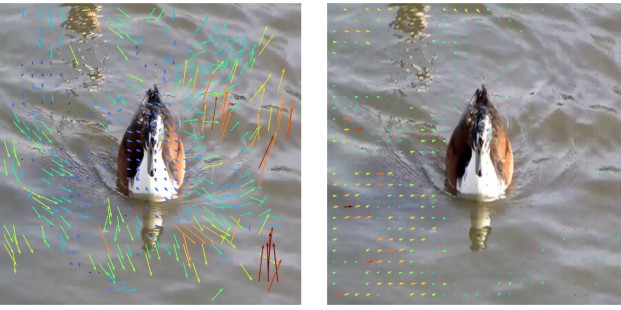

"The snake is heading towards the right side of the frame."

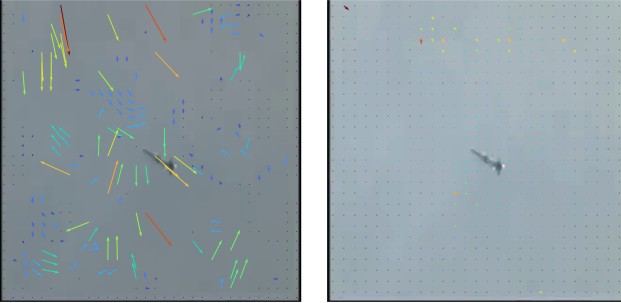

"The duck is moving forward in the water."

"The missile is heading towards the ground."

Figure 4: **Failure cases (Tracker).** Left – noisy displacement vectors from CoTracker3. Right – corresponding displacement vectors between adjacent frame latents produced by our method on smolVLM. Below each pair appears the caption generated by smolVLM, illustrating poor motion understanding caused by these noisy latents.

In many scenes, object motion is primarily induced by camera movement rather than the object's own displacement, as the camera is fixed on the object. In Fig. 4, we see that misunderstandings of background displacement vectors, stemming from limitations of the tracker, cause the model to infer incorrect motion directions.

