# OpenReview forum: "VideoMB: Steering Representations towards Motion Balanced Caption Generation in Vision-Language Models"
_ICLR.cc/2026/Conference — ICLR 2026 Conference Desk Rejected Submission_

### Official Review · Reviewer_6hXR · 2025-10-25

**Soundness:** 2
**Presentation:** 2
**Contribution:** 2
**Rating:** 4
**Confidence:** 3

**Summary:**

This paper introduces VideoMB, a framework designed to correct the appearance bias in Vision-Language Models (VLMs) and improve their understanding of motion. The method incorporates cross-attention layers between consecutive video frames to enhance features of moving objects. It is fine-tuned using a dual objective: the standard captioning loss and a novel auxiliary loss, which uses an off-the-shelf point tracker to supervise the learning of spatiotemporal correspondences. Experiments demonstrate that this lightweight approach significantly improves motion-based captioning on two different VLMs, particularly in challenging scenarios with small or blurry objects.

**Strengths:**

Originality: The approach of using an external point tracker to generate supervision for a patch-matching loss is a highly novel and effective way to explicitly inject motion priors into a VLM.

Quality: The technical execution is strong, with a well-designed method validated by comprehensive experiments. The consistent improvements on two different VLM architectures and standard benchmarks, supported by insightful qualitative visualizations, are convincing.

Clarity: The paper is exceptionally well-written and easy to follow. The problem, method, and results are presented logically and clearly, with effective diagrams and figures.

Significance: The work provides a practical and computationally efficient solution to a fundamental limitation in video understanding. Its plug-and-play nature makes it a valuable tool for improving a wide range of VLMs.

**Weaknesses:**

1. Dependency on External Tracker: The method's performance is inherently tied to the quality of the external point tracker (CoTracker). The paper does not sufficiently discuss the impact of tracker failures or noise, which could lead to the model learning flawed motion patterns.
2. Unquantified Inference Overhead: While the fine-tuning is efficient, the paper claims the overall approach is computationally efficient without quantifying the added latency or computational cost (e.g., FLOPs) of the new cross-attention layers during inference.
3. Limited Temporal Scope: The model only processes relationships between consecutive frames. This may be insufficient for understanding long-range temporal dependencies or events where key actions occur across non-adjacent frames.

**Questions:**

1. How robust is VideoMB to noise or errors from the CoTracker? Have you analyzed how performance degrades when the tracker fails on challenging videos (e.g., with severe occlusions or non-rigid objects)?
2. Could you please quantify the inference overhead introduced by VideoMB? For instance, what is the percentage increase in latency or FLOPs compared to the baseline models?
3. The current design focuses on consecutive frames. How does this limit the model's ability to understand long-range temporal relationships, and have you considered strategies to extend it to non-adjacent frames?

---

> ### Author Response · Authors · 2025-11-22
>
> We thank the reviewer for the constructive feedback and provide answers below.
>
> > W1&Q1. Dependency on External Tracker… the paper does not sufficiently discuss the impact of tracker failures
>
> We acknowledge that analyzing failure cases arising from the external tracker’s limitations is important. The core limitation of CoTracker, and consequently of our method, is its difficulty in tracking points on featureless surfaces (e.g., sky, water), where it produces noisy displacement vectors. This directly affects our model’s ability to capture motion patterns in such scenarios. We have prepared qualitative examples illustrating these failure cases and their impact on our model’s learning, which are presented in Appendix A of the revised manuscript.
>
> > W2 & Q2. Unquantified Inference Overhead: While the fine-tuning is efficient, the paper claims the overall approach is computationally efficient without quantifying the added latency or computational cost (e.g., FLOPs) of the new cross-attention layers during inference.
>
> We thank the reviewer for this valuable feedback. We provide the quantified inference overhead results in Table A below.
>
> **Table A:** Inference overhead comparison between the baseline models and our method, measured by FLOPs and inference latency, averaged over 1500 samples taken from the Tempcomp dataset.
> | Model                | Version     | FLOPs (T) | Latency (ms) |
> |----------------------|-------------|-----------|---------------|
> | SmolVLM          | Baseline    |    27.395     |   1552.466 |
> |                      | Finetuned   |    28.706     |    1567.169|
> | InternVL         | Baseline    |   24.515     |     232.77 |
> |                      | Finetuned   |    26.225     |    247.315|
>
>
> We observe a modest FLOPs increase of 4.7% for SmolVLM and 6.9% for InternVL, and a latency increase of 1% for SmolVLM and 6.4% for InternVL.
>
> > W3 & Q3. Limited Temporal Scope: The model only processes relationships between consecutive frames. This may be insufficient for understanding long-range temporal dependencies or events where key actions occur across non-adjacent frames.
>
>  Long-range temporal dependencies are what the autoregressive text-space reasoning of the pretrained model already handles effectively. We are not replacing this long-range temporal reasoning but rather providing better motion-aware visual inputs for it.
> Moreover, while we directly align only consecutive frames, the cumulative effect of stacked cross-attention layers creates an effective temporal receptive field that extends beyond adjacent frames.
>
> Finally, our strong performance across well-adopted motion-centric benchmarks demonstrates that consecutive frame alignment is sufficient for the motion understanding tasks we target. If long-range non-adjacent frame relationships were critical for these benchmarks, we would observe performance degradation.

---

> > ### Author Response · Authors · 2025-11-27
> >
> > Dear Reviewer 6hXR,
> >
> > Thank you for taking the time to review our paper.
> >
> > As the discussion period comes to a close, we kindly ask whether you have any remaining concerns.
> >
> > In particular, we believe that our responses have fully addressed the following points:
> >
> > W.1: Discussion on the limitations of CoTracker and how its failure modes affect our method.
> >
> > W.2: Quantification of computational overhead introduced by our method.
> >
> > W.3: Clarification that our method enhances motion-aware visual inputs rather than replacing long-range temporal reasoning, and that our empirical results demonstrate that consecutive frame alignment suffices for our targeted motion understanding tasks.
> >
> > If you have any remaining concerns, please let us know. We would be happy to provide further clarification.

---

### Official Review · Reviewer_MTgH · 2025-10-30

**Soundness:** 2
**Presentation:** 2
**Contribution:** 1
**Rating:** 2
**Confidence:** 5

**Summary:**

This manuscript introduces VideoMB, a novel framework designed to address the inherent appearance bias in Large Vision-Language Models (VLMs) when applied to video captioning. The authors identify a core problem: the tokenization process in VLMs compresses visual information, leading to a loss of temporal fidelity and a consequent failure to adequately capture and describe moving objects. VideoMB tackles this by integrating two key components into pre-trained VLMs: 1) Cross-attention layers between consecutive frames to enhance temporal alignment and feature matching, and 2) An auxiliary loss function that enforces similarity between corresponding patch embeddings in adjacent frames, using point tracks from an off-the-shelf model (CoTracker) as supervision.

**Strengths:**

1. The paper clearly identifies a significant and under-explored problem in VLMs: the trade-off between spatial fidelity and temporal reasoning leading to motion neglect.
2. A key strength of VideoMB is its practical design, particularly for analyzing the motion of small objects in VLMs.

**Weaknesses:**

1. The architecture relies on established techniques: cross-attention for feature alignment and self-supervised losses for temporal consistency. Since these are well-known in video understanding, the overall novelty of the model is constrained.
2. The method's auxiliary loss relies on pseudo-labels from CoTracker. This introduces a dependency on the performance and biases of this external model.
3. The cross-attention mechanism is exclusively applied between consecutive frames. This design may struggle to capture long-range temporal dependencies or motions. The model might miss patterns that require a broader temporal context for accurate reasoning.
4. The model encourages feature similarity for corresponding patches but does not explicitly represent or output motion features (like optical flow or displacement vectors) that could be directly utilized by the LLM backbone. The motion understanding remains an implicit property of the embeddings, which may limit the LLM's ability to perform explicit spatio-temporal reasoning.
5. The ablation study, while present, is somewhat limited. It primarily ablates the loss function components but does not ablate the core architectural change, which is the cross-attention layers themselves. Questions remain: How much performance gain comes from the cross-attention alone versus the auxiliary loss? What is the effect of the number and placement of these layers?
6. The experimental evaluation primarily demonstrates the improvement achieved by applying VideoMB to two specific base models (InternVL3-1B and smolVLM2). However, it lacks a comprehensive comparison against other recent and state-of-the-art (SOTA) video understanding VLMs that are specifically designed for temporal reasoning, such as MotionSight, CogVLM2, or VideoLLaMA. Without this broader comparison, it is difficult for the reader to assess whether VideoMB establishes a new SOTA on these benchmarks or simply narrows the performance gap between efficient base models and more sophisticated, temporally-aware architectures.

**Questions:**

1. Could you please specify the exact number ("K") and the insertion points of the cross-attention layers within the vision encoder (e.g., after which transformer blocks)? What was the rationale for this specific configuration?
2. What is the individual contribution of the cross-attention mechanism versus the auxiliary similarity loss? Have you experimented with using the cross-attention layers but with only the standard captioning loss
3. Was there a specific rationale for choosing a point tracker like CoTracker over more traditional, dense optical flow methods? Given that your auxiliary loss requires point-level correspondences, did you consider deriving pseudo-tracks by sampling points and then tracking them using a dense flow estimator?
4. Was there a specific reason you chose to use cross-attention layers rather than, for instance, applying a Parameter-Efficient Fine-Tuning (PEFT) technique like LoRA to the existing self-attention or feed-forward layers within the pre-trained vision encoder?

---

> ### Author Response · Authors · 2025-11-25
>
> We thank the reviewer for their constructive feedback and provide answers below.
>
> > W1. The architecture relies on established techniques: cross-attention for feature alignment and self-supervised losses for temporal consistency … the overall novelty of the model is constrained.
>
> We respectfully disagree with this characterization. To the best of our knowledge, there is currently no work that incorporates either a cross-attention mechanism or a self-supervised auxiliary loss specifically to promote motion reasoning in vision–language models (VLMs). Our method is the first to propose a generic, low-level framework to enhance visual representations of moving elements.
>
> Our novelty lies not just in applying those techniques, but also in demonstrating why and how feature-level temporal alignment addresses an existing core architectural limitation. Concretely, we: (1) leverage motion-based point tracking to generate patch-level supervision that explicitly reinforces moving representations, and (2) introduce cross-attention along the vision pathway before text-space projection, enabling superior temporal reasoning where it was previously absent, as validated by our empirical results.
>
> > W2. The method's auxiliary loss relies on pseudo-labels .... This introduces a dependency on the performance and biases of this external model.
>
> Reliance on external models for pseudo-labels is common in prior influential works and an accepted practice in the field [1,2]. We acknowledge that analyzing how failure cases of such models affect ours is important.
>
> The core limitation of CoTracker, and consequently of our method, is its difficulty in tracking points on featureless surfaces (e.g., sky, water), where it produces noisy displacement vectors. This directly affects our model’s ability to capture motion patterns in such scenarios. We have prepared qualitative examples illustrating these failure cases and their impact on our model’s learning, which are presented in Appendix A of the revised manuscript.
>
> > W3. The cross-attention is applied between consecutive frames … This design may struggle to capture long-range temporal dependencies
>
> Our strong performance across well-adopted motion-centric benchmarks demonstrates that consecutive frame alignment is sufficient for the motion understanding tasks we target. If long-range non-adjacent frame relationships were critical for these benchmarks, we would have observed performance degradation.
>
>  Long-range temporal dependencies are what the autoregressive text-space reasoning of the pretrained model already handles effectively. We are not replacing this long-range temporal reasoning but rather providing better motion-aware visual inputs for it.
>
> Moreover, while we directly align only consecutive frames, the cumulative effect of stacked cross-attention layers creates an effective temporal receptive field that extends beyond adjacent frames.
>
> > W4. The model… does not explicitly represent or output motion features (like optical flow or displacement vectors) that could be directly utilized by the LLM backbone. The motion understanding remains an implicit property of the embeddings, which may limit the LLM's ability to perform explicit spatio-temporal reasoning.
>
> The reviewer assumes that explicit motion features are a prerequisite for temporal understanding, but our results demonstrate otherwise. While adding explicit representations might offer further improvements, this would require expanding the LLM's input modality, and in general, this research direction is beyond our scope. Our work shows that implicit motion encodings are sufficient to improve temporal reasoning. Introducing explicit representations would change the research question from "can we improve existing architectures?" to "can we redesign the input space?"
>
> ***
>
> **References:**
>
> [1] Tracktention: Leveraging Point Tracking to Attend Videos Faster and Better. lai et al. CVPR 2025
>
> [2] VideoJAM: Joint Appearance-Motion Representations for Enhanced Motion Generation in Video Models. Chefer et al. ICML 2025

---

> ### Author Response · Authors · 2025-11-25
>
> > W5 & Q2. The ablation study is somewhat limited. Questions remain: How much performance gain comes from the cross-attention alone versus the auxiliary loss?
>
> We thank the reviewer for this valuable suggestion and agree that it is crucial to establish that without the auxiliary loss, we would not be able to incorporate motion priors into the model. As suggested by the reviewer, we’ve expanded our ablation study to include these results, as well as results using different weighting values for the auxiliary loss. Our findings are presented in Table 3 in our revised manuscript.
>
> As shown in Table 3, ablating the cross-alignment loss results in substantially degraded performance compared to our full method. This is expected: fine-tuning solely on the captioning objective does not inherently encourage the cross-attention mechanism to capture inter-frame differences, nor to enhance representations of moving elements.
>
> > Q1. Could you please specify the exact number ("K") and the insertion points of the cross-attention layers within the vision encoder (e.g., after which transformer blocks)? What was the rationale for this specific configuration?
>
> We selected 𝐾=4 cross-attention layers for all models, uniformly distributed across the vision encoder, to balance representational coverage and computational feasibility. This configuration provided a practical trade-off - capturing temporal cues at multiple abstraction levels without substantially increasing training cost. In terms of computational overhead, this configuration allows for a modest FLOPs increase of 4.7% for SmolVLM and 6.9% for InternVL, and a latency increase of 1% for SmolVLM and 6.4% for InternVL.
>
> > Q3. Was there a specific rationale for choosing a point tracker like CoTracker over more traditional, dense optical flow methods? Given that your auxiliary loss requires point-level correspondences, did you consider deriving pseudo-tracks by sampling points and then tracking them using a dense flow estimator?
>
> VLMs sample discrete frames from continuous video sequences. This requires understanding information flow across temporally distant frames–something most optical flow methods and dense flow estimators are not adjusted for. Moreover, CoTracker produces binary visibility masks indicating whether each tracked point is visible, and our ablation study shows that incorporating these masks during fine-tuning is critical for improved performance.
>
> > Q4. Was there a specific reason you chose to use cross-attention layers rather than, for instance, applying a Parameter-Efficient Fine-Tuning (PEFT) technique like LoRA to the existing self-attention or feed-forward layers within the pre-trained vision encoder?
>
> In our setting, training the existing self-attention layers, even with PEFT methods like LoRA, would not achieve the behavior we require because these layers process each frame independently. They cannot propagate information across frames, and this propagation is required for our method. Cross-attention enables this cross-frame flow of information.

---

> ### Author Response · Authors · 2025-11-27
>
> Dear Reviewer MTgH,
>
> Thank you for taking the time to review our paper.
>
> As the discussion period comes to a close, we kindly ask whether you have any remaining concerns.
>
> In particular, we believe that our responses have fully addressed the following points:
>
> W.1: We clarified that our novelty lies in demonstrating why and how feature-level temporal alignment addresses motion understanding at a low level, with empirical validation of this approach's effectiveness.
>
> W.2: We explained that relying on external models is a common approach and added a discussion on the limitations of CoTracker and how its failure modes affect our method.
>
> W.3: We clarified that our method is used to enhance motion-aware visual inputs, not to replace long-range temporal reasoning, and that our empirical results over well-adopted benchmarks demonstrate that consecutive frame alignment suffices for our targeted motion understanding tasks.
>
> W.4: We clarified that incorporating explicit representations is not a prerequisite for enhanced temporal understanding, and that our results demonstrate the sufficiency of implicit motion encodings.
>
> W.5: We expanded our ablation study and demonstrated that ablating the cross-alignment loss substantially degrades performance.
>
> Q1-Q4: We explained the clear rationale for every architectural design choice included in our method.
>
> If you have any remaining concerns, please let us know. We would be happy to provide further clarification.

---

### Official Review · Reviewer_Uj6n · 2025-10-30

**Soundness:** 2
**Presentation:** 2
**Contribution:** 3
**Rating:** 2
**Confidence:** 3

**Summary:**

This paper showed VLMs have bias towards appearance, where models often fail to capture motion dynamics in videos, and instead describing a summary of static objects. The authors introduce VideoMB, a fine-tuning framework designed to steer a VLM's representations towards temporal reasoning. The core of the method consists of two main contributions: (1) integrating new cross-attention layers between consecutive frame embeddings to model temporal information flow, and (2) introducing a dual-objective loss function. This loss combines the standard captioning objective with a novel motion-similarity loss​. This auxiliary loss is supervised by displacement vectors extracted from an off-the-shelf point tracker (CoTracker), encouraging the model's internal representations to align with real-world motion patterns. The framework is applied to two existing VLMs (SmolVLM2 and InternVL3-1B), showing improved performance on motion-centric video understanding benchmarks.

**Strengths:**

- The core idea of using an external, high-quality point tracker to generate a supervisory signal for a VLM's internal representations is solid.
- By only fine-tuning the newly introduced cross-attention layers and keeping the base VLM's weights frozen , the approach is computationally efficient and can be seamlessly integrated into existing pre-trained models.

**Weaknesses:**

- The paper fails to evaluate the trade-offs of improving motion understanding. The entire objective is to steer the model's representations towards motion, which could plausibly harm its ability to perform appearance-focused tasks.
- The authors themselves note that the evaluation benchmarks (TempCompass and MotionBench) primarily test "fine-grained human movements" with "limited motion magnitude". This seems misaligned with the method's potential.
- Insufficient architectural details: the provided "details" are very limited and I have concern on reproducibility.

**Questions:**

The weaknesses are significant. The complete dependency on an external tool without a thorough analysis of its failure modes, the lack of evaluation on static tasks to check for negative transfer, and the somewhat limited scope of the quantitative evaluation prevent me from recommending acceptance at this time.

- Please add a discussion on the limitations of CoTracker and how its failure modes could impact VideoMB. (A qualitative example showing a case of tracker failure and its effect on the model's learning would be highly informative.)
- Evaluating the fine-tuned models on a standard static image captioning benchmark (like COCO) and an appearance-centric video task to show that you are not harming the model's general abilities.
- Please provide a deeper analysis of the poor performance on the "Order" sub-task in TempComp.

---

> ### Author Response · Authors · 2025-11-22
>
> We thank the reviewer for their detailed comments and suggestions and provide responses to the raised concerns below.
> > W1 & Q2. The paper fails to evaluate the trade-offs of improving motion understanding … could plausibly harm its ability to perform appearance-focused tasks.
>
> Our framework intentionally specializes the model for motion-centric tasks, similar to how Meta's recent work [1] on depth prediction fine-tuning evaluated only depth tasks, without assessing general vision capabilities. When adapting VLMs for specialized domains, some capability trade-offs are expected and acceptable. In our case, we acknowledge that this trade-off allows for superior motion understanding, as validated by our empirical results, at the cost of appearance-focused capabilities.
>
> > W2. … evaluation benchmarks primarily test "fine-grained human movements" with "limited motion magnitude". This seems misaligned with the method's potential.
>
> We respectfully disagree that this constitutes a weakness of our work. First, we demonstrate significant improvements on the available benchmarks, showing that our method enhances temporal understanding even within their scope. The gains we observe on fine-grained movements underscore the robustness of our approach: if our motion-aware objective improves performance on subtle temporal changes, this suggests even stronger benefits for more dynamic scenarios, as supported by our qualitative results.
>
> > W3. Insufficient architectural details: the provided "details" are very limited…
>
> We respectfully disagree with the reviewer’s assessment. The architectural details and fine-tuning procedure are thoroughly described at the beginning of Section 4, providing sufficient information to reproduce our method. Moreover, our code is available. Nonetheless, we are happy to provide additional implementation details if the reviewer believes specific aspects are unclear or missing.
>
> >Q1. Add a discussion on the limitations of CoTracker and how its failure modes could impact VideoMB.
>
> We thank the reviewer for this valuable suggestion and agree that discussing CoTracker's limitations is crucial. The core limitation of CoTracker, and consequently our method, is its struggle to track points on featureless surfaces (e.g., sky, water), where it produces noisy displacement vectors. This directly impacts our model's ability to understand motion patterns in such scenarios. As suggested by the reviewer, we have prepared qualitative examples demonstrating these failure cases and their effect on our model's learning, presented in Appendix A in our revised manuscript.
>
> > Q3. Please provide a deeper analysis of the poor performance on the "Order" sub-task in TempComp
>
> As the reviewer rightfully noted, both models evaluated in our paper exhibit markedly poor performance on the ‘Order’ sub-task in TempComp. While we did address this issue in our original manuscript, we agree that a more careful and comprehensive analysis remains important.
>
> Upon closer inspection among the samples that the fine-tuned variants misclassified while their baseline counterparts predicted correctly,  we find that these samples predominantly fall into two main categories: (1) videos with minimal motion cues, and (2) videos featuring non-naturalistic scenarios, including physically implausible movements or abrupt scene discontinuities. This is further compounded by the fact that the 'Order' sub-task questions are typically framed as ‘What was the order of events?’ without explicitly grounding these events to specific objects. As a result, the model may overly emphasize events whose motion patterns better align with its learned distribution.
>
> We note that a similar metric, ‘Action order’, which measures the model’s understanding of the order in which events occur, is evaluated in the MotionBench benchmark. In this metric, we achieve a substantial improvement, noting that the MotionBench samples consist exclusively of videos depicting natural, continuous movements with clear motion cues, precisely the conditions under which our motion-focused fine-tuning demonstrates its strengths.
>
> ***
> **References:**
>
> [1] DepthLM: Metric Depth From Vision Language Models. Cai et al. Meta.

---

> > ### Author Response · Authors · 2025-11-27
> >
> > Dear Reviewer Uj6n,
> >
> > Thank you for taking the time to review our paper.
> >
> > As the discussion period comes to a close, we kindly ask whether you have any remaining concerns.
> >
> > In particular, we believe that our responses have fully addressed the key aspects of your review. For example:
> >
> > W.1: We clarified that adapting the model for enhanced motion understanding results in an intentional trade-off over appearance-biased capabilities, which is acceptable and common when adapting VLMs for specialized tasks.
> >
> >
> > W.2: We elaborated on how the significant improvements achieved on the available benchmarks demonstrate the effectiveness of our method.
> >
> > Q.1: We added a discussion on the limitations of CoTracker and how its failure modes affect our method.
> >
> > Q.3: We provided a deeper analysis of why the performance was lower specifically on the "Order" metric in TempComp.
> >
> > If you have any remaining concerns, we would be happy to clarify further.

---

> ### Comment · Reviewer_Uj6n · 2025-11-28
> **Response to the Authors**
>
> Thank you for your response.
>
> W1. I personally oppose to this view but I think the argument is reasonable.
>
> W2. Which result is the significant improvement that you are referring to?
>
> W3. I think the section 3 is hard to follow, and even if the code is available, it is better to clarify the method in the main article I believe.
>
> Q1. Thank you for providing the failure modes from CoTracker in the revised appendix.
>
> Q3. I don't think "substantial improvement"is presented in MotionBench for Action Order.
>
> I am keeping my original score for now.

---

> > ### Author Response · Authors · 2025-11-28
> >
> > We thank the reviewer for their response, and the opportunity to respond.
> >
> > W2: We refer the reviewer to our results in Tables 2-3 for TempComp and MotionBench benchmarks, respectively. Notably, on TempComp, our method achieves 4%+ gains in ‘Direction Understanding’ and ‘Attribute Change’ detection for SmolVLM2. On MotionBench, we observe even stronger performance: +12% (SmolVLM2) and +16% (InternVL3) for ‘Repetition Count’, +6% (SmolVLM2) and +1% (InternVL3) for ‘Action Order’, and +2% to +3% for ‘Motion Recognition’, with consistent gains across most remaining motion-related tasks.
> >
> > The fact that we achieve measurable improvements on benchmarks that primarily comprise subtle, fine-grained movements—even though our method is designed to enhance motion comprehension more broadly—strongly supports that our approach addresses a fundamental limitation in current VLMs' temporal understanding.
> >
> > W3: Could the reviewer please identify which specific aspects of Section 3 are unclear or limited? We are happy to add clarifications, but currently find the description comprehensive.
> >
> > Q3: We respectfully disagree with this assessment. Our method achieves +6% for SmolVLM2 and +1% for InternVL3 on the 'Action Order' metric in MotionBench (Table 3). We argue these improvements are substantial given that: (1) the gains are consistent across two architecturally distinct models, indicating a robust, generalizable effect rather than noise; (2) they were achieved without any additional training data; and (3) the computational overhead is negligible. In video-language understanding, we maintain that such consistent gains achieved under such constraints represent meaningful progress in motion comprehension.

---

### Official Review · Reviewer_U4Gk · 2025-10-31

**Soundness:** 3
**Presentation:** 3
**Contribution:** 2
**Rating:** 4
**Confidence:** 2

**Summary:**

The paper presents a new vision language model which provides a better captioning taking into account not just the frame features but also the fluent information across the frames. Using only the frame features often leads to loss of temporal information. This is addressed by incorporating a consecutive frame cross attention layer which captures correlation of information across the evolving frames. The loss function is accordingly modified to to minimise both feature representation error as well as temporal misalignments. The approach is likely to work for loss of temporal information in the form of occlusion and noise. Extensive experimental results are presented for various kinds of video scenes and noise conditions. The proposed method shows promising results.

**Strengths:**

1) The cross-attention layer across frames is a novel approach.
2) The loss function is suitable for the proposed task.
3) Handles occlusion and noise conditions
4) extensive experimental results are presented

**Weaknesses:**

1) Using only consecutive frame alignment via cross attention might be simplistic for complex video. Many temporal events requiring semantic segmentation might be undetected.
2) There might be overfitting in the case of pretraining and fine tuning in case of OOD videos.
3) Comparison with many state of art video captioning LLM is missing
4) Ablation study should include hyeprparameter tuning effects like loss function compositions under varying weighting schmes etc

**Questions:**

1) What is the effect of transition across semantic video segments on the cross attention layer modeling consecutive frame alignment? Are these effects smoothened out. Are the caption generation features not so much dependent on segment transition?

2) Comparison with other transition aware captioning schemes should be made. See for example:
Progress-Aware Video Frame Captioning, Zihui Xue, Joungbin An, Xitong Yang, Kristen Grauman; Proceedings of the IEEE/CVF Conference on Computer Vision and Pattern Recognition (CVPR), 2025

3) The ablation study may be extended by considering various hyper-parameter combinations.

---

> ### Comment · Reviewer_U4Gk · 2025-11-26
>
> We appreciate some of the ablation studies that are now pointed to. Some of the arguments regarding auxiliary knowledge is still missing.

---

> ### Author Response · Authors · 2025-11-27
>
> We are grateful for the reviewer’s feedback and provide answers below.
>
> > W1 & Q1. Using only consecutive frame alignment via cross attention might be simplistic for complex video. Many temporal events requiring semantic segmentation might be undetected.
>
> We thank the reviewer for raising this concern. Our evaluation benchmark, TempComp, already consists of samples with multiple video segments. Our strong performance across this well-adopted benchmark suggests that our cross-attention mechanism does not pose a limitation for semantic segmentation reasoning, and that consecutive-frame alignment is sufficient for the motion-understanding tasks we target.
>
> We acknowledge that some trade-off exists for the model’s reasoning capabilities about the order of events taking place in each segment. This is partially why we observe poorer performance specifically for the ‘Order’ metric in the TempComp benchmark for both models evaluated, while achieving substantial improvement on a similar metric, ‘Action Order’, in the MotionBench benchmark. MotionBench samples consist exclusively of videos depicting natural, continuous movements with clear motion cues—precisely the conditions under which our motion-focused fine-tuning demonstrates its strengths.
>
> > W2. There might be overfitting in the case of pretraining and fine tuning in case of OOD videos
>
> We perform fine-tuning solely on the GOT-10K dataset [1], which includes over 10,000 videos covering a wide range of object types and diverse motion patterns and scales. Our empirical results which are evaluated on the TempComp and MotionBench benchmarks, show significant performance gains, suggesting that no overfitting occurred; otherwise, we would have expected a degradation in performance. We underline that the fine-tuning and evaluation are carried out on different datasets.
>
> > W4 & Q3. Ablation study should include hyeprparameter tuning effects
>
> We thank the reviewer for this valuable suggestion. We’ve expanded our ablation study to include these results. Our findings are presented in Table 3 in our revised manuscript.
>
> As shown in Table 3, we observe minor variations in performance across the range of auxiliary loss weightings evaluated, with a general trend of decline as the weighting value increases. Overall, performance remains stable, demonstrating the robustness of our method. We note a slightly larger shift exclusively for the 'Repetition Count' metric on MotionBench – this metric may be more sensitive to weighting changes as it evaluates samples that are less well-aligned with the learned distribution. Nonetheless, for all weighting values, we consistently match or outperform the baseline for this metric.
>
> > Q2 Comparison with other transition aware captioning schemes should be made.
>
>  We respectfully disagree that our work requires comparison to transition-aware captioning schemes such as the one presented in ProgressCaptioner [2], as they address a fundamentally different problem and operate at different stages of the video understanding pipeline.
>
> ProgressCaptioner focuses on fine-grained event captioning, achieved by breaking down sequences of frames, whereas our work addresses a lower-level representation learning problem across the entire temporal dimension: reducing the model's bias toward static visual features at the frame-encoding stage, which prevents effective capture of motion dynamics.
> Furthermore, ProgressCaptioner evaluates primarily on event-focused captioning benchmarks, whereas our improvements target motion-understanding capabilities.
>
>
> ***
>
> **References:**
>
> [1] Got-10k: A large high-diversity benchmark for generic object tracking in the wild. Huang et al. IEEE 2021
>
> [2] Progress-Aware Video Frame Captioning. Xue et al.  Proceedings of the IEEE/CVF Conference on Computer Vision and Pattern Recognition (CVPR), 2025

---

### Note · Program_Chairs · 2026-01-17
**Submission Desk Rejected by Program Chairs**

The following references in this submission do not refer to real documents and/or have major errors in bibliographic information:

 Jiapeng Wang, Jian Jiao, Yunchao Wei, and Yi Yang. Vthcl: Video temporal hard contrastive learning. In Proceedings of the 30th ACM International Conference on Multimedia (ACM MM), pp. 4793-4801, 2022